

# Daily hypoxia forecasting and uncertainty assessment via Bayesian mechanistic model for the Northern Gulf of Mexico

Alexey Katin[1], Dario Del Giudice[1], Daniel R. Obenour[1]

[1]Department of Civil, Construction, & Environmental Engineering, North Carolina State University, Raleigh,
27606, USA

*Correspondence to*: Alexey Katin (akatin@ncsu.edu)

**Abstract.** Low bottom water dissolved oxygen conditions (hypoxia) occur almost every summer in the northern
Gulf of Mexico due to a combination of nutrient loadings and water column stratification. Several models have
been used to forecast the midsummer hypoxic area based on spring nitrogen loading from major rivers. However,
sub-seasonal forecasts are needed to fully characterize the dynamics of hypoxia over the summer season, which
is important for informing fisheries and ecosystem management. Here, we present an approach to forecast hypoxic
conditions at daily resolution through Bayesian mechanistic modeling that allows for rigorous uncertainty
quantification. Within this framework, we develop and test different representations and projections of hydro-
meteorological model inputs. We find that May precipitation over the Mississippi River Basin is a key predictor
of summer discharge and loading that substantially improves forecast performance. Accounting for spring wind
conditions also improves forecast performance, though to a lesser extent. The proposed approach generates
forecasts for two different sections of the Louisiana–Texas shelf (east and west), and it explains about 50% of the
variability in total hypoxic area when tested against historical observations (1985−2016). Results also show how
forecast uncertainties build over the summer season, with longer lead times from the nominal forecast release date
of 31 May, due to increasing stochasticity in riverine and meteorological inputs. Consequently, the portion of
overall forecast variance associated with uncertainties in data inputs increases from 26% to 41% from June–July
to August–September, respectively. Overall, the study demonstrates a unique approach to assessing and reducing
uncertainties in dynamic hypoxia forecasting.

## Introduction

The Northern Gulf of Mexico (NGoM) has one of the largest hypoxic zones in the world, forming virtually every
summer over the last three decades (Rabalais and Turner, 2019). Hypoxic or "dead" zones occur when dissolved
oxygen concentrations fall below critical thresholds (e.g., 2 mg/L) threatening aquatic ecosystems (Craig, 2012;
Craig and Crowder, 2005; Thronson and Quigg, 2008), fisheries (Purcell et al., 2017; Smith et al., 2017), and
coastal economies (Díaz and Rosenberg, 2011). Two major causes of hypoxia in the Gulf are water column
stratification and nutrient loadings (Krug, 2007; Obenour et al., 2012; Rabalais et al., 2002), which are both
influenced by Mississippi and Atchafalaya River (MAR) discharges. Additionally, wind controls both the
structure of the river plume (Hetland, 2005) and the rates of oxygen supply to the water column (Fennel et al.,
2013; Justić et al., 1996). Overall, a complex combination of biophysical factors including long-term accumulation
of organic matter (Del Giudice et al., 2020; Turner et al., 2008) and short-term events like storms and droughts
(Bianchi et al., 2010) control hypoxia dynamics in the NGoM.



Mathematical models are useful for elucidating important relationships between hypoxia and environmental drivers, and for evaluating the consequences of possible actions to improve water quality (Justić and Rose, 2017). The approaches developed to predict hypoxia in the Gulf of Mexico included statistical regressions (Forrest et al., 2011; Greene et al., 2009; Turner et al., 2012), as well as both parsimonious (Obenour et al., 2015; Scavia et al., 40 2013) and complex (Justić and Wang, 2014; Yu et al., 2015) mechanistic models. Among these alternatives, parsimonious process-based models attempt to balance biophysical detail with computational efficiency and resilience to overfitting. When embedded in a Bayesian framework, these models describe eutrophication processes and hypoxia formation while enabling data-driven parameter estimation and rigorous uncertainty analysis (Ménesguen and Lacroix, 2018). The latter is especially important for assessing our confidence in the 45 potential outcomes of environmental change and management decisions (Reichert and Borsuk, 2005; Schuwirth et al., 2019). Currently, a probabilistic ensemble of four models is used to inform stakeholders and fishery managers about the expected extent of the NGoM hypoxic zone (Scavia et al., 2017). This ensemble provides predictions with estimates of uncertainty of the midsummer hypoxic area (HA) in the NGoM. However, the forecast lacks dynamic oxygen predictions over the summer season. The lack of subseasonal information on 50 dissolved oxygen variability has been identified as an important limitation in understanding how hypoxia affects fisheries in the region, which occur primarily during the summer season but are highly dynamic in space and time (Langseth et al., 2016; Purcell et al., 2017; Smith et al., 2014). Laurent and Fennel (2019) used a weighted aggregation of seasonal hindcasts generated by a three-dimensional model to produce spatially and temporarily resolved seasonal hypoxia forecasts, but without accounting for uncertainties related to the model 55 parameterization. Further, the aforementioned forecasting approaches are informed only by observed spring nutrient loading, without considering variability in spring wind conditions (Obenour et al., 2015) or projected summer MAR discharge and loading.

Here, we use a Bayesian mechanistic model to forecast the temporal dynamics of hypoxia, with a focus on how different model inputs influence forecast accuracy and uncertainty. The model was initially developed by Obenour 60 et al. (2015) and later enhanced by Del Giudice et al. (2020) (hereafter referred to as DMO20). While the model performed well in hindcasting, its ability to forecast hypoxia forward in time has not been explored. In order to provide sufficient lead for environmental planning and fisheries management, we produce a June–September hypoxia forecast based on data available at the end of May. The main objectives of this study are to: a) develop daily forecasted spatial-mean bottom water dissolved oxygen (BWDO) concentrations and HA estimates for 65 targeted portions of the Louisiana–Texas Shelf with accompanying measures of uncertainty; b) understand the major sources of forecast uncertainty; and c) characterize how forecast accuracy degrades over time; and d) explore how different applications of spring-summer riverine and meteorological data influence forecast performance.

## 2 Methods

We first outline the underlying model and required data inputs. Next, we describe the proposed forecasting procedure, along with regression models to project discharge and nitrogen loading over the summer. Third, we describe the approach to evaluate BWDO and HA forecast performance and analyse how the forecast varies in relation to alternative combinations of data inputs.



### 2.1 Bayesian mechanistic model and bias adjustment

The BWDO forecast is based on the model described in DMO20, which has a parsimonious mechanistic formulation and coarse spatial resolution. Specifically, the model represents the Louisiana–Texas shelf from Galveston Bay to the Mississippi River Delta, divided into west and east sections at the Atchafalaya River mouth (Fig. S1). The water column of each sections is divided by the pycnocline into two layers, assuming that discharge and nutrient loadings are transported within the top layer. Additionally, wind speed and direction control the

distribution of flow and loadings between the east and west sections as well as the rate of reoxygenation across the pycnocline. The biogeochemistry is based on the transformation of bioavailable nitrogen (sum of nitrate, nitrite, ammonia, and 12% organic nitrogen (Obenour et al., 2015)) into organic matter, which settles to the bottom layer and is subject to aerobic decomposition. BWDO is depleted due to both near- and long-term oxygen demands, reflecting the effects of nitrogen loadings over different time scales. Therefore, the model uses both

recent inputs of daily discharge, loading, and wind (up to 90 d before the date of prediction) and long-term November–March loading. The model predicts BWDO, which is then transformed into HA via regressions. The Bayesian calibration framework provides systematic estimation of model parameters and their uncertainties (DMO20).

Prior to developing the daily forecast, we examined DMO20 predictions for systematic biases during specific

months and found that hindcasted BWDO was (0–20%) lower than observations for the west section of the shelf in June (Section S2). This apparent bias in the DMO20 model was corrected using a linear regression, with the day number (June 1 to June 30) as a predictor and the BWDO adjustment factor as response (Fig. S2.1). This adjustment was applied to all June model predictions unless otherwise indicated.

### 2.2 Data

The forecast utilized the same observational data inputs described in DMO20, including monthly discharge and nitrogen loading from the U.S. Geological Survey (USGS, 2019), daily discharge from the U.S. Army Corps of Engineers at Simmesport and Tarbert Landing (USACE, 2019), and wind speed and zonal velocities from a National Data Buoy Center (NDBC, 2019). Additionally, monthly precipitation and temperature were obtained from gridded data for the MAR Basin (Hart and Bell, 2015; Schwartz, 2012). Section-specific observations of

BWDO and HA with associated uncertainties used for DMO20 calibration were obtained via averaging the geostatistical estimates from Matli et al. (2018). Similar to DMO20, we only used the geostatistical space–time estimates corresponding to times of monitoring cruises, as these estimates generally have lower uncertainties.

### 2.3 Forecast procedure

To capture the uncertainty in hydrometeorology, nutrient loading, model parameters, and residual error, the

forecast for a given year was determined through 1000 Monte Carlo simulations (Fig. 1). Each simulation included a random draw from the Bayesian joint posterior parameter and error distribution of the mechanistic model (DMO20), and the uncertainty in the regression to convert BWDO to HA. The simulations used actual November–May riverine and meteorological inputs for the forecast year, since these inputs would be known by the nominal forecast release date of 31 May. However, summer (June–September) inputs were sampled from historical

records. To retain the temporal correlation in these inputs, data were sampled in blocks (i.e., the complete daily





time series for each summer). As wind conditions cannot be accurately forecasted beyond 10 days (Zhang et al., 2019), summer wind velocity data were sampled from the complete historical record (1985–2016).

Summer riverine inputs can potentially be projected in advance through regression (Section 2.4). Thus, riverine time series were sampled from only the 10 most "relevant" historical years. For each historical year, relevancy to

the forecast year was determined by computing the differences between the summer historical records and the regression-projected discharge and bioavailable nitrogen loading. Both monthly projections and observations were standardized based on the mean and standard deviation of the historical data for each summer month (1980– 2016), so that the differences in loading and flow could be combined on the same unitless scale. The 10 years with the smallest aggregated differences were selected as the relevant years for use in the Monte Carlo simulations.


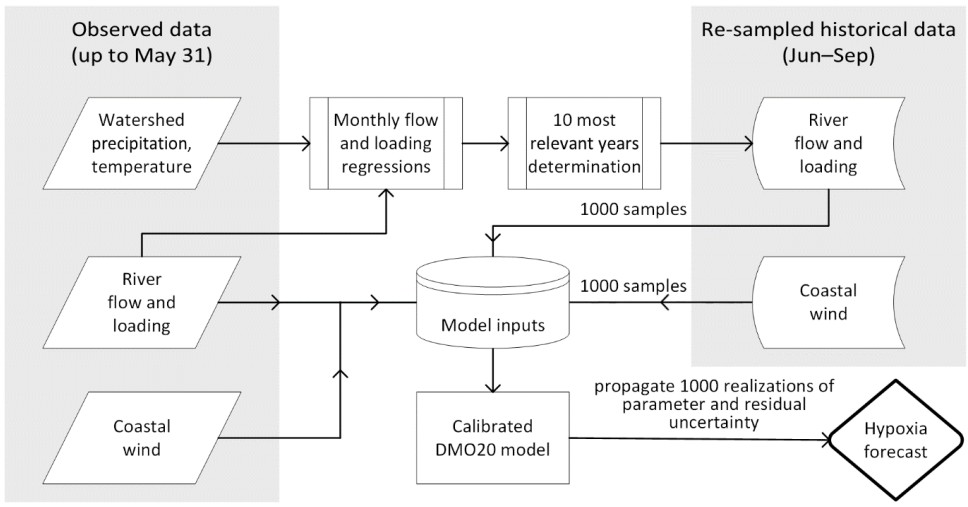

**Figure 1: Flowchart summarizing processes and data required to generate the hypoxia forecast for a given year.**

**2.4 Regressions for June-September discharge and loading**

Regression modeling was used to project riverine inputs, which were employed to constrain the historical records used in the Monte Carlo simulations to relevant years (Section 2.3, Fig. 1). June, July, August, and September MAR discharge ($Q_A$ and $Q_M$, m$^3$/s) and bioavailable nitrogen loading ($L_A$ and $L_M$, T/mo) were estimated through multiple linear regression. The candidate predictor variables (predictors) included monthly (January–May) and 4 month average (January–April) discharge, loading, total river basin precipitation ($P$, in), and river basin

temperature ($T$, ºC) (Fig. S3.1). Response variables were square-root transformed to account for the skewness of their distributions and comply with error normality assumption for linear regression (Faraway, 2015). Predictors for each model were selected using the Bayesian Information Criterion (BIC) through an exhaustive search (Lumley, 2017). BIC prioritizes models based on log likelihood while penalizing for the number of parameters to prevent overfitting (Faraway, 2015). The performance of the regression was measured by the coefficient of

determination, $R^2$, in the square-root transformed space. If any of the sixteen regressions had $R^2 \leq 30\%$, the





associated month and variable was excluded from determining relevant years for hypoxia forecasting (Section 2.3).

### 2.5 Forecast assessment

The forecasting approach was applied to the historical record (1985–2016), while excluding the summer input
data of the forecast year from the Monte Carlo simulation. This retrospective forecast (i.e., pseudo-forecast) performance was evaluated through comparison of the daily forecasted values with both hindcasted (generated by DMO20) and geostatistically estimated (referred to as "observed" for brevity) BWDO and HA for the two shelf sections. The approach also allowed for determining 95% inter-quantile ranges (IQR) of the pseudo-forecasts, encompassing variability in summer riverine and meteorological inputs, parameter uncertainty, and residual error
(mechanistic model residual and transformation of BWDO to HA error).

We also assessed how inclusion of various hydrometeorological inputs affected pseudo-forecast accuracy and uncertainty. Specifically, we compared four cases with different types of spring–summer wind and summer riverine data. Case 1 included summer riverine and spring–summer wind records randomly sampled from the complete historical data (thus they are treated as unknown, consistent with conventional Gulf forecasting
approaches). Case 2 was similar to Case 1, except it included actual spring wind data (to 31 May). Case 3 was also similar to Case 1, except it used summer riverine records sampled from only the 10 most relevant historical years, as determined from the regression projections (Section 2.4). Finally, Case 4 (our proposed approach, Section 2.3, Fig. 1) used both actual spring wind data and riverine records from the 10 most relevant years.

### 3 Results and Discussion

#### 3.1 Monthly discharge and loading projections

Sixteen multiple linear regressions predict average monthly summer discharge and bioavailable nitrogen loading for the MAR. The performance of the regressions generally decreases from the beginning to the end of summer (Table 1), reflecting the increasing temporal gap (i.e., lead time) between the available spring predictors and the forecast response. For instance, the regressions explain 78% and 9% of the variability in (square-root transformed)
Mississippi River bioavailable nitrogen loading in June and September, respectively. The residuals for all selected models appear evenly distributed with minimal heteroscedasticity (Fig. S3.2–S3.5) and mostly weak serial correlation of residuals (Pearson lag 1 correlation ($r$) ranging from –0.02 to 0.35). The predictive variables chosen via exhaustive BIC selection include May discharge from both rivers ($Q_{A5}$ and $Q_{M5}$) or bioavailable nitrogen loading ($L_{A5}$ and $L_{M5}$) in 13 of the 16 models. In other words, high flow and nutrient loading in May is indicative
of high flow and nutrient loading in summer. However, the most consistent individual predictor (present in 12 out of 16 regressions) is MAR Basin precipitation in May ($P_5$), likely due to the hydrologic lag between rainfall and basin discharge. Note that the correlation between $P_5$ and May discharge is relatively weak ($r = 0.36$ for both rivers), while the correlations between $P_5$ with June and July discharge are $r = 0.77$ and $r = 0.66$, respectively, suggesting an average basin response time of 1–2 months. This lag is generally consistent with a previous study
that identified a strong positive correlation between March–May precipitation and May–June nitrogen flux in the basin (Donner and Scavia, 2007). Additionally, the strong influence of lagged watershed precipitation on nitrogen loading has been confirmed for other river basins (Gentry et al., 2014; Hinsby et al., 2012; Sinha and Michalak,





2016). The eight regressions for June and July discharge and bioavailable nitrogen loading are used for screening and constraining riverine inputs for subsequent hypoxia forecasting (Section 2.3). However, regressions for

August and September are excluded from the screening due to relatively poor performance ($R^2 \leq 0.3$), compared to the earlier months ($R^2 \geq 0.45$).

**Table 1: Regressions for monthly Atchafalaya and Mississippi River discharge ($Q_A$ and $Q_M$, m³/s) and bioavailable nitrogen loading ($L_A$ and $L_M$, Mg/mo). Variable subscript numbers represent months. For example, $P_{1:4}$ represents**
**MAR average basin precipitation for January-April. Bold $R^2$ values (>0.30) indicate models used for selection of relevant years in hypoxia forecasting.**

|  | Regression | $R^2$ |
|---|---|---|
| Atchafalaya | $\sqrt{Q_{A6}} = 19.45 + 2.55 \times 10^{-3} \times Q_{A5} + 0.46 \times P_5$ | **0.79** |
|  | $\sqrt{Q_{A7}} = 8.20 + 0.54 \times P_{1:4} + 0.38 \times P_5$ | **0.47** |
|  | $\sqrt{Q_{A8}} = 33.69 + 1.06 \times 10^{-3} \times Q_{A5} + 0.19 \times P_5$ | 0.28 |
|  | $\sqrt{Q_{A9}} = 63.15 + 2.61 \times T_{1:4}$ | 0.13 |
|  | $\sqrt{L_{A6}} = 35.86 + 1.57 \times 10^{-3} \times L_{A5} + 0.82 \times P_5$ | **0.76** |
|  | $\sqrt{L_{A7}} = 54.44 + 1.17 \times 10^{-3} \times L_{A5} + 0.54 \times P_5$ | **0.51** |
|  | $\sqrt{L_{A8}} = 72.82 + 1.22 \times 10^{-3} \times L_{A5}$ | 0.30 |
|  | $\sqrt{L_{A9}} = 73.61 + 0.56 \times 10^{-3} \times L_{A5}$ | 0.11 |
| Mississippi | $\sqrt{Q_{M6}} = 31.63 + 1.66 \times 10^{-3} \times Q_{M5} + 0.69 \times P_5$ | **0.77** |
|  | $\sqrt{Q_{M7}} = 49.01 + 0.73 \times 10^{-3} \times Q_{M5} + 0.51 \times P_5$ | **0.48** |
|  | $\sqrt{Q_{M8}} = 53.11 + 0.69 \times 10^{-3} \times Q_{M5} + 0.28 \times P_5$ | 0.28 |
|  | $\sqrt{Q_{M9}} = 97.19 + 3.81 \times T_{1:4}$ | 0.13 |
|  | $\sqrt{L_{M6}} = 32.42 + 0.92 \times 10^{-3} \times L_{M5} + 1.66 \times P_5$ | **0.78** |
|  | $\sqrt{L_{M7}} = 44.85 + 0.65 \times 10^{-3} \times L_{M5} + 1.39 \times P_5$ | **0.51** |
|  | $\sqrt{L_{M8}} = 50.87 + 0.52 \times 10^{-3} \times L_{M5} + 0.80 \times P_5$ | 0.25 |
|  | $\sqrt{L_{M9}} = 92.39 + 1.57 \times P_5$ | 0.09 |

### 3.2 Forecast skill

After constraining historical riverine inputs (to the 10 most relevant years) based on the discharge and loading
regressions for each forecast year, the hypoxia model is run to obtain daily hypoxia predictions (Fig. 2, top). Over the 32 year record, these pseudo-forecasts explain 66% and 64% of the variability in hindcasted BWDO (i.e., DMO20 model predictions assuming all inputs are known throughout the summer) for the west and east sections, respectively. After transformation of BWDO to HA, the pseudo-forecast explains 68% of variability in hindcasted HA for each section (Fig. S4.1). Overall, the pseudo-forecast explains 71% and 77% of the variability in
hindcasted total shelfwide HA and mean BWDO, respectively.

Pseudo-forecasts can also be compared to observed (geostatistically estimated) BWDO and HA at the times of monitoring cruises (Fig. 2, bottom). The forecasted BWDO fits moderately well to the observations with an $R^2$ of 0.39 and 0.50 for the west and east sections, respectively. When BWDO is transformed to HA, the pseudo-forecast



explains 41% and 48% of variability in observed HA in the west and east sections, respectively (Fig. S4.1), which

is similar to the hindcast explanatory power of 46% and 58%, (as in DMO20). In comparison, hindcasting studies using three-dimensional models have generally explained a lower 27–37% of the variability in Gulf BWDO, but at finer spatial resolution (Fennel et al., 2016). To our knowledge, this is the first time Gulf hypoxia forecasts have been rigorously compared to observations across the entire summer season (June–September). Previous studies have generally focused on assessing forecast performance relative to the Louisiana Universities Marine

Consortium midsummer shelf-wide hypoxia cruises, which typically takes place within a two-week window beginning in late July (Laurent and Fennel, 2019; Scavia et al., 2017).

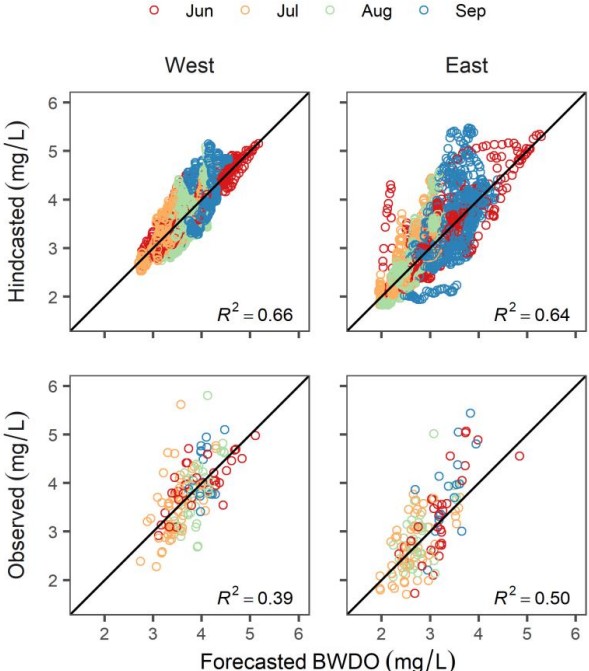

**Figure 2: Daily hindcasted (available for every summer day from DMO20) and observed (geostatistically estimated**
**from monitoring cruises) BWDO versus pseudo-forecasted BWDO for west and east sections. Diagonal line represents perfect prediction.**

Forecasting performance gradually degrades with longer lead times. The ability of pseudo-forecasts to match DMO20 hindcasts of shelfwide HA declines by about 50% (comparing $R^2$ values, Fig. 3, top) from June to

September due to increasing uncertainty in riverine and meteorological inputs toward the end of the summer season. In comparison, the ability of forecasts to match actual observations declines by nearly 70% (Fig. 3, bottom). Forecasts and hindcasts benefit from the same seasonal patterns inherent to the DMO20 model structure, while observations may deviate from these patterns due to additional drivers of variability not captured in the mechanistic formulation.




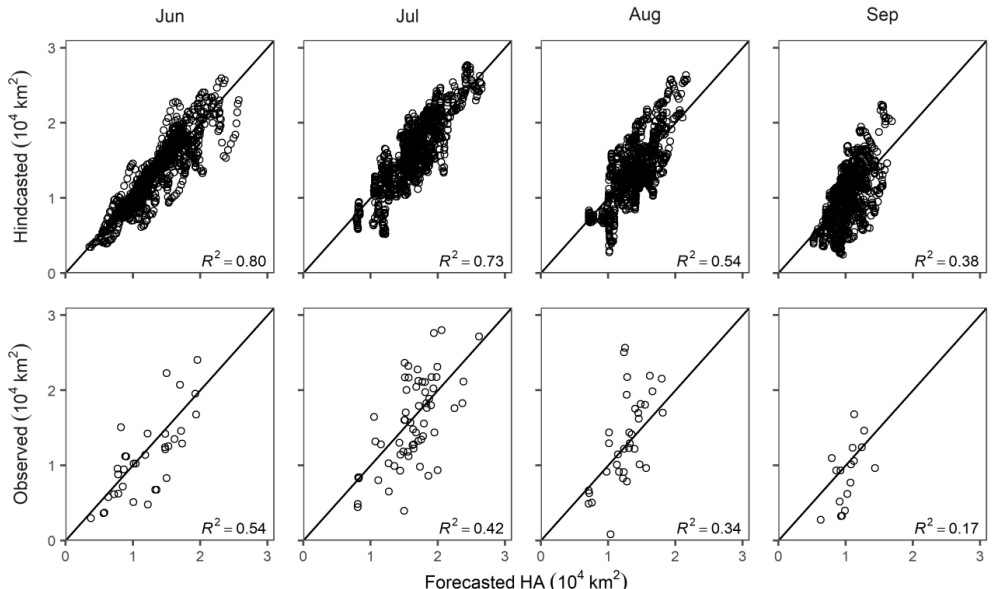

**Figure 3: Month-by-month comparison of the daily hindcasted (top) and observed (geostatistically estimated, bottom) HA versus pseudo-forecasted HA. Diagonal line represents perfect prediction.**

Our forecast quantifies predictive uncertainty associated with the Bayesian model parameter estimates, variability in riverine and meteorological inputs, and residual error (Fig. 4, Figs. S5.1–S5.11). The results indicate that 95% IQR for the west section is on average 2.6 times higher than for the east section, due to greater overall size of the west section (and greater HA) and the complex effect of both river outfalls (Atchafalaya and Mississippi) on BWDO in this section (DMO20). Although the forecasts generally follow the shape of the hindcasts over time,

some dissimilarities exist due to hydrometeorological variability (Fig. 4). The pseudo-forecast captures the large HA during summer 1993 that was caused by extremely high May–September MAR flow and nutrient loadings (Larson, 1997). Interestingly, however, the pseudo-forecast in 2009 overpredicts HA in the west section (two observations are outside of the 95% IQR), due to unusually strong westerly summer winds in this year (Turner et al., 2012). Generally, high wind stress increases water column reaeration and disrupts stratification (Justić and

Wang, 2014; Obenour et al., 2015), while upwelling westerly winds disperse the river plume offshore, reducing the consequent oxygen demand (Feng et al., 2012; Le et al., 2016). Overall, only 6.7% (10 of 149) of the observations of total HA are outside of the 95% IQR (Figs. S6.1–S6.8). Also, the geostatistically estimated 95% confidence intervals of observed HA always overlap the forecasted 95% IQR except for one observational cruise in 1988. This discrepancy is caused by anomalously strong summer winds combined with low discharge and

nutrient loading in 1988 (Tables S3.1–S3.2). In general, these results suggest the forecasts realistically characterize uncertainties.

Our forecasting approach allows for the contribution of uncertainty in parameters, data inputs, and residual error to be disentangled. Note that the relative magnitudes of the variance components are somewhat different from the relative magnitudes of the 95% IQR components (e.g., Fig. 4), as the IQR is related to standard deviation (i.e.,



square root of variance), which is not readily separable. For total HA, the variance associated with riverine and meteorological inputs is 40 times greater than variance associated with parameter uncertainty (on average). On the other hand, the variance associated with residual error is 1.7 times greater than the variance related to the data inputs. Also, the variance associated with data inputs is more influential in later months, as its portion of the overall forecast variance increases from 26% in June–July to 41% in August–September. Furthermore, residual uncertainty is heavily influenced by the transformation of BWDO to HA (Section 2.1), and exhibits higher variance with larger HA (Fig. S4.2). The relatively low parameter uncertainty reflects the long calibration record available (currently 1985–2016) and is consistent with the underlying model's robust performance in cross validation (Obenour et al., 2015).

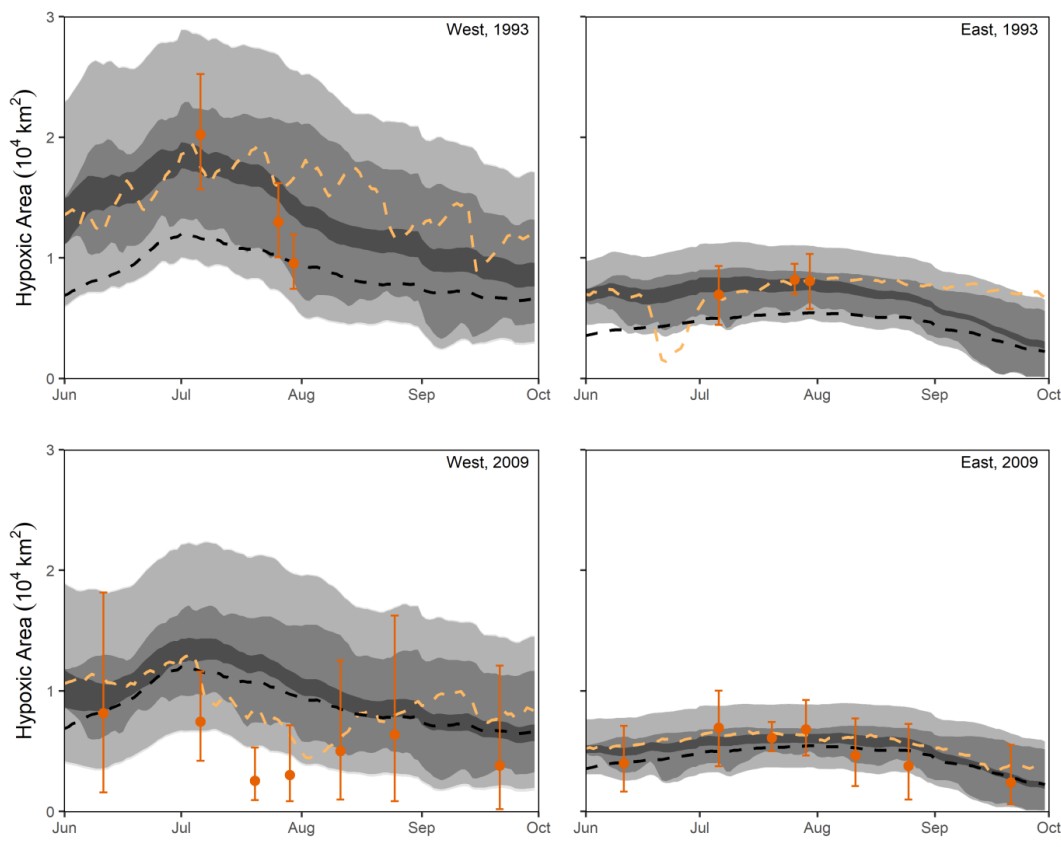

**Figure 4: Daily pseudo-forecasts of HA for the west and east sections in 1993 (top) and 2009 (bottom), including 95% IQR of the predictive distribution, distinguishing between i) parameter, ii) riverine and meteorological inputs, iii) mechanistic model error and iv) regression related to transformation of BWDO to HA uncertainties (shades of gray from darkest to lightest). Yellow dashed line is hindcasted estimate, black dashed line is the 32 year average hindcast, orange points and error bars represent the mean and associated 95% confidence interval of the (geostatistically estimated) hypoxia observations.**

The forecasting uncertainties vary throughout the summer, mostly due to the transition from observed to randomly sampled model inputs. The riverine and meteorological inputs to DMO20 are lagged rolling window averages that



include mostly actual observed data (i.e., data prior to 1 June) at the beginning of the summer, and an increasing

proportion of randomly sampled historical data thereafter (Section 2.1). As a result, the normalized IQR (i.e., IQR divided by predicted HA) for pseudo-forecasted HA for June–July is 30% lower than for August–September on average (Fig. 5A, boxplots). Interestingly, there is a rapid increase in normalized IQR during the second week of June. In the model, mean water column reaeration is determined by wind speeds over the preceding two weeks (Obenour et al., 2015). Consequently, the large and highly variable wind speeds of June (Fig. 5B) (de Velasco

and Winant, 1996) quickly increase predictive uncertainties, as these stochastic inputs replace the known wind speed inputs from May. Finally, increases in IQR are also noted in August and September (Fig. 5), which is consistent with limited ability of the regression analysis (Section 3.1) to accurately project flow and load for such long lead times.

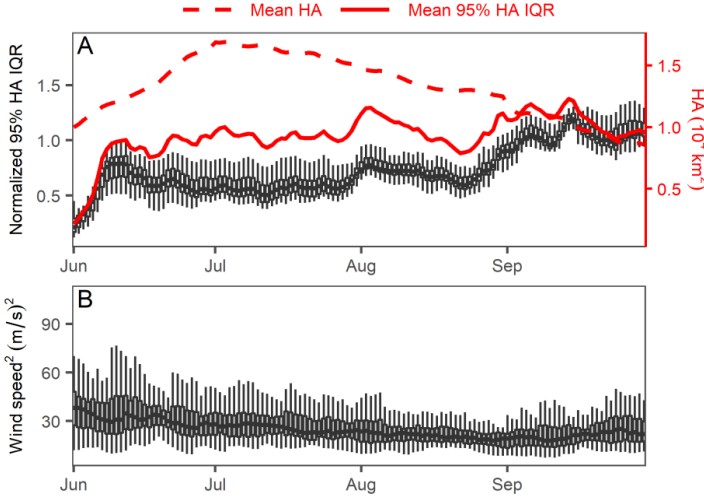


**Figure 5: (A) Boxplots represent daily pseudo-forecasted normalized IQR (IQR/HA) including uncertainty due to variability in parameters, riverine inputs and meteorology. Red dashed and solid lines show the daily mean HA and (non-normalized) IQR, respectively; (B) Boxplots show 14 day weighted average squared wind speed near the west shelf section. Boxplots represent interannual variability in the results. The center of each box is the median, while**

**whiskers extend to the extreme value or 1.5 times the IQR of the corresponding variable (whichever is less).**

### 3.3 Sensitivity to riverine and meteorological inputs

The results presented in the previous section are for the proposed forecasting approach with known spring loadings, discharge and winds, and with summer riverine inputs constrained through the regression projections

(i.e., Case 4 of Section 2.5). In comparison, the more conventional Gulf forecasting approach, using known spring riverine inputs but with unknown wind and summer riverine inputs (i.e., randomly sampled from the entire historical record, Case 1) explains only around half of variability in hindcasted and observed HA (i.e., 56% and 44%, respectively, Table 2). Also, the performance of Case 1 declines greatly from the beginning (June) to the end (September) of the summer. The inclusion of summer riverine records constrained through regression

projections substantially increases the variance explained in both hindcasted and observed HA (Table 2,





Cases 3 and 4). This improvement in performance is the most notable for July–September, indicating that the constraining summer riverine historical records provides a more accurate determination of water column stratification and oxygen demand within the biophysical model. Additionally, the summer-wide average of the normalized IQR for Case 4 is on average 22% lower than that of the conventional forecasting approach (Case 1).

Addition of actual spring wind data to the conventional approach (Table 2, Case 2) slightly increases the explained variance in hindcasted and observed HA by 4% (i.e., from 56% to 60%) and 1%, respectively. This forecast improvement is most notable in June and July because zonal wind velocities up to three months in advance regulate the transport of water and nutrients over the shelf (Obenour et al., 2015; Walker et al., 2005). Interestingly, about a quarter of the variability in hindcasted June HA remains unexplained even when actual

spring wind data is included (Table 2, Case 2). This is consistent with the importance of near-term wind and discharge inputs in controlling reaeration in the model. It is also consistent with the uncertainties presented in Fig. 5 and other modeling studies exploring the influence of wind on hypoxia formation (Forrest et al., 2011; Yu et al., 2015). Overall, the forecast is only moderately sensitive to the inclusion of actual spring wind velocities (compare Cases 1 and 2 to Cases 3 and 4). However, we anticipate that inclusion of actual spring wind data may

still be important, especially for years with anomalous wind patterns.

Finally, for the preferred forecasting approach (Case 4) we examine an alternative way of determining the relevant years that constrain the distribution of riverine inputs for the forecast year. If only nitrogen loading regressions are used to constrain summer inputs (instead of both flow and loading regressions), the explained variability in the hindcasted total HA drops from 71% (Table 2, Case 4) to 69%. This relatively small drop in predictive

performance is not too surprising, as monthly nitrogen loading, which is the primary driver of many hypoxia models (Turner et al., 2006), is highly correlated with monthly discharge ($r = 0.90$). However, employing the discharge regressions (in addition to the loading regressions) better accounts for the influence of river flow on stratification and hypoxia formation (Obenour et al., 2012).





**Table 2: Explained variance ($R^2$) in hindcasted and geostatistically observed HA by pseudo-forecasts based on the different data input cases. Here, *act* indicates actual data for the specific year being forecasted are used, *ran-all* indicates input data are randomly sampled from the complete historical record, and *ran-sel* indicates that riverine input data are randomly sampled from a subset of relevant historical years based on the regression projections for flow and loading. Spring includes March–May while summer includes June–September records.**

| Case | Data input | | | | $R^2$, Forecasted vs Hindcasted | | | | | $R^2$, Forecasted vs Observed | | | | |
| --- | --- | --- | --- | --- | --- | --- | --- | --- | --- | --- | --- | --- | --- | --- |
| | Riverine | | Wind | | Month | | | | Overall | Month | | | | Overall |
| | Spring | Summer | Spring | Summer | Jun | Jul | Aug | Sep | | Jun | Jul | Aug | Sep | |
| 1 | *act* | *ran-all* | *ran-all* | *ran-all* | 0.69 | 0.38 | 0.32 | 0.21 | 0.56 | 0.57 | 0.28 | 0.28 | 0.15 | 0.44 |
| 2 | *act* | *ran-all* | *act* | *ran-all* | 0.76 | 0.56 | 0.33 | 0.19 | 0.60 | 0.53 | 0.32 | 0.28 | 0.14 | 0.45 |
| 3 | *act* | *ran-sel* | *ran-all* | *ran-all* | 0.73 | 0.68 | 0.52 | 0.36 | 0.68 | **0.58** | 0.39 | 0.32 | **0.19** | 0.49 |
| 4 | *act* | *ran-sel* | *act* | *ran-all* | **0.80** | **0.73** | **0.54** | **0.38** | **0.71** | 0.54 | **0.42** | **0.34** | 0.17 | **0.50** |


### 3.4 Implications for hypoxia forecasting and fisheries management

To our knowledge, there is only one hypoxia forecasting study (i.e. Laurent and Fennel, 2019), with a similar temporal scope to the current study. That study applied three-dimensional hydrodynamic–biogeochemical model hindcasts, weighted based on comparisons with historical May nitrogen loading only. Other predictive hypoxia

studies have employed both discharge and wind (Forrest et al., 2011; Testa et al., 2017), however they lacked the desired temporal resolution of this study. Here, we demonstrate how projections of summer riverine inputs based on spring discharge, loading, and watershed precipitation (Section 3.1) can be used to constrain model inputs, substantially improving hypoxia forecasting skill (Table 2). We suggest that other hypoxia forecasting efforts could also benefit from such expanded and projected model inputs.

Our approach allows for daily forecasts of BWDO and HA for two sections of the NGoM throughout the entire summer season. Generally, results indicate that HA can be forecasted up to four months ahead, but predictions for later months should be treated with increased caution given their higher uncertainties. Note that the MAR regressions only explain 25–30% of the variability in August nitrogen loadings (Table 1), which is a major factor underlying the decrease in forecast skill in late summer. Additional sources of uncertainty in model inputs may

arise from extreme climatic events (e.g. tropical storms and hurricanes), which are unpredictable at the forecasting time scales considered here. These storm events have a multifaceted effect on BWDO, potentially reaerating the bottom water column but also providing additional terrestrially derived or sediment-resuspended nutrients that may exacerbate hypoxia (Bianucci et al., 2018; Yu et al., 2015). Therefore, it is not surprising that the pseudo-forecast can explain only about a third of the variability in hindcasted September total HA (Fig. 3). Future hypoxia

forecasting efforts would benefit from improvements in weather and riverine forecasting systems that provide more reliable projections for longer time periods.

The forecasts also explicitly distinguish between different sources of uncertainty in BWDO and HA (Fig. 4). Most previous forecasting studies for the NGoM (Forrest et al., 2011; Scavia et al., 2013; Turner et al., 2012) and other systems like Chesapeake Bay (Testa et al., 2017) implicitly represent uncertainty associated with unknown

summer data inputs, sometimes accounting for it in the residual error. On the other hand, the recent study by





Laurent and Fennel (2019) explicitly considers uncertainty due to stochastic summer riverine inputs, but does not include parameter and residual error uncertainties in the generated forecasts. Thus, this study provides a more comprehensive uncertainty assessment, allowing for management decisions that are robust to potential extremes (Keeney, 1982; Schuwirth et al., 2019).

Finally, the proposed approach has potential benefits for short- and long-term environmental planning and fisheries management. In the NGoM, the largest volume (Atlantic Menhaden) and highest valued (Penaeid shrimp) fisheries occur during the summer months concurrent with seasonal hypoxia. These fisheries are highly mobile and hypoxia is known to affect the dynamics of both targeted species (Craig and Crowder, 2005; Craig and Bosman, 2013) and fishing fleets (Langseth et al., 2014; Purcell et al., 2017), with potential implications for catch

(Craig, 2012), economic condition (Smith et al., 2017), and management (Langseth et al., 2016). However, previous attempts to correlate fishery performance (e.g. catch) with annual measures of hypoxic severity (e.g., area of hypoxia in late July) have had limited success (O'Connor and Whitall, 2007; Zimmerman and Nance, 2001) because neither the spatio–temporal dynamics of hypoxia or of the fishery have been considered. Thus, the proposed daily forecasts can potentially be linked to fisheries and ecosystem models (e.g. de Mutsert et al., 2016),

to provide more actionable management guidance.

## 4 Conclusion

In this study, we demonstrate a novel approach for forecasting intra-seasonal variability in BWDO and HA in the NGoM by leveraging a Bayesian mechanistic model. This study generates the first daily hypoxia forecasts across the summer season (up to four months ahead) with comprehensive uncertainty assessment. We show that the

major sources of uncertainty include variability in data inputs and residual error, while model parameter uncertainty is relatively small. This study also compares how different methods for specifying riverine and meteorological model inputs influence forecast accuracy. In particular, we show how constraining summer riverine inputs based on spring conditions, including precipitation over the MAR Basin, can be used to improve hypoxia forecasting skill. We also show that inclusion of monitored spring wind data further improves hypoxia

forecasts. Together, these enhancements increase retrospective pseudo-forecast accuracy from 44% to 50% ($R^2$) while reducing forecast uncertainty by 22% across summers, relative to the conventional approach using spring loadings and flows only (with randomly sampled summer riverine and spring–summer wind records). Thus, the forecasting system developed here provides an enhanced capacity to inform natural resources management in hypoxic coastal systems.

**Acknowledgements**

We thank Dr. Kevin Craig for a detailed review of the manuscript. This work was funded by the NOAA grant NA16NOS4780203.

**Code/Data Availability**

Hydrometeorological and water quality data can be obtained from the public sources described in the methods. A
complete set of compiled input data can be provided upon request (akatin@ncsu.edu).





**Author Contribution**

All authors designed the research, analysed the results and prepared the manuscript. Alexey Katin developed the codes and created tables and figures.

**Competing interest**

The authors declare that they have no conflict of interest.

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
