# Peer review of "Temporally resolved coastal hypoxia forecasting and uncertainty assessment via Bayesian mechanistic modelling"

_Hydrology and Earth System Sciences, 2021_

## Author Comment (AC1)

**Responses to comments on: "Daily hypoxia forecasting and uncertainty assessment via Bayesian mechanistic model for the Northern Gulf of Mexico" (Referee #1)**

Our responses are in blue.

**General comments**

One important piece of information that is not mentioned explicitly enough in the introduction or the abstract is that the main part of the manuscript is about using a statistical model to generate suitable parameters for an existing mechanistic model (referred to DMO20). Reading these sections for the second time, it becomes a bit more clear but it would be beneficial to the reader to describe this more clearly early in the manuscript.

Thank you for your feedback. We will clarify in the introduction that in this manuscript we are using an existing mechanistic model and associated posterior parameter distributions, which were determined through Bayesian inference.

The authors are careful in creating a forecasting scenario in which no summer data is included. However, the entire set of historical data is used to produce the linear regressions, which may include future data w.r.t. year for which the forecast is produced.

We appreciate the comment and understand the concern of the referee. Note that we do not use the predictions of these regressions directly in the forecast; we only apply them for defining the 10 relevant years. To double-check the validity of these regressions, we performed leave-one year-out cross validation (LOOCV), where we excluded years by one, calibrated the models to the reduced dataset and predicted for the excluded year. The performance of August–September dropped dramatically for cross-validated results compared to models built on the full dataset. However, as highlighted in the Results (Lines 173-175), we are only using the June–July regressions for relevant years selection. We will add the description of this additional analysis to the manuscript and expand Table 1.

| Model | $R^2$ from Table 1 | LOOCV $R^2$ | % Change |
|---|---|---|---|
| $\sqrt{Q_{A6}}$ | 0.79 | 0.75 | -5.1 |
| $\sqrt{Q_{A7}}$ | 0.47 | 0.38 | -19.1 |
| $\sqrt{Q_{A8}}$ | 0.28 | 0.14 | -50.0 |
| $\sqrt{Q_{A9}}$ | 0.13 | 0.04 | -69.2 |
| $\sqrt{L_{A6}}$ | 0.76 | 0.71 | -6.6 |
| $\sqrt{L_{A7}}$ | 0.51 | 0.41 | -19.6 |
| $\sqrt{L_{A8}}$ | 0.3 | 0.17 | -43.3 |
| $\sqrt{L_{A9}}$ | 0.11 | -0.02 | -118.2 |
| $\sqrt{Q_{M6}}$ | 0.77 | 0.73 | -5.2 |
| $\sqrt{Q_{M7}}$ | 0.48 | 0.38 | -20.8 |
| $\sqrt{Q_{M8}}$ | 0.28 | 0.15 | -46.4 |

| | | | |
|---|---|---|---|
| $\sqrt{Q_{M9}}$ | 0.13 | 0.04 | -69.2 |
| $\sqrt{L_{M6}}$ | 0.78 | 0.74 | -5.1 |
| $\sqrt{L_{M7}}$ | 0.51 | 0.41 | -19.6 |
| $\sqrt{L_{M8}}$ | 0.25 | 0.12 | -52.0 |
| $\sqrt{L_{M9}}$ | 0.09 | 0.01 | -88.9 |

If I understand the approach correctly, even in the Case 4 setup, the forecast may include information from summer data of the current year, if the current year is "relevant" (as defined in the manuscript). As a result, does the forecasting system produce significantly better results for the "relevant" years compared to other years? In addition to the 4 cases currently included in the manuscript, I would suggest to add a Case 5 that excludes all data from the future (forecasts for the first few years with little data could be skipped) or, alternatively, excludes the data from the current forecast year, even if it is relevant.

Actually, the pseudo-forecasts do not include the information from the summer of the current year (forecast year) in any of the forecasting scenarios, as stated in the Methods at Lines 139-140. In other words, when the procedure selects the 10 relevant years, the summer data for the forecast year is excluded from selection. We will add a brief note to the Results (Section 3.2) to remind readers of this important point. We do not think that the suggested Case 5 is warranted for this reason and because it does not directly address the study's research objectives (Lines 63-68).

We will modify Figure 1 (see draft below) to clarify that the forecasts only use observed data up until 31 May of the forecast year (since the nominal forecast release date is 1 June). Data for after 31 May are sampled only from "other years".

Looking at the results in Fig. 2 and 3, it appears as if the forecast-hindcast as well as the forecast-observation comparisons show a pattern in August and September: The forecast appears to overestimate BWDO and HA for low values and underestimate it for high values and this pattern appears to increase in time. The authors already introduce a linear regression for the purpose of bias reduction but apply it only to the June forecast. A similar linear correction could be applied to correct this pattern which appears to increase with lead time. Yet, I am a bit hesitant to recommend such a correction because it, just like the bias reduction, adds a non-mechanistic element to the model.

Thank you for the comment, in fact we agree with the reviewer regarding some bias presence in September for the east section. However, we decided not to introduce a September bias correction for two main reasons. First, there are fewer measurements of hypoxia available for September, so the case for a bias correction is less strong (Fig. S2.2). Second, we see that HA is predicted surprisingly well in September, even without a bias correction (Fig. S.3). We also note that hypoxia is typically reduced in September compared to June–August, so September is of less interest to management and fisheries compared to other months.

**(A) Hindcast**

[Figure]

**(B) Pseudo–forecast**

[Figure]

How difficult would it be to extend the approach presented in this study and estimate August and September values from all data available until then? I am not suggesting that this needs to be done in this manuscript, yet creating successive 2 month forecasts appear a suitable course of action for producing more accurate estimates. This could be mentioned in the discussion.

We appreciate the comment and agree that being able to update and extend the forecast over the course of the summer could be beneficial for fisheries management. In this manuscript, we focus on a 1 June forecast release date, consisting with current Gulf forecasting practices. However, the methods should be transferable to other forecast release dates. We will expand the Discussion, pointing this out as a future opportunity.

Overall, while the manuscript is well written it sometimes overestimates the study-specific knowledge of the reader. Including more information explicitly would benefit readers. In some instances, I had to read ahead to answer questions which could have been addressed right away. I have listed some of those instances below.

Thank you for the positive feedback. In addition to addressing your specific comments, we will review and edit the entire manuscript for clarity.

**Specific comments**

L 8: "Several models" Here it would be helpful to specify what type of model is meant, e.g. "dynamical", "statistical" etc.

We appreciate the comment and will add clarification to the abstract. Furthermore, we expand on the modeling approaches to predict and forecast hypoxia later in introduction (Lines 38-40, 46-55).

L 52: The same group has previously considered different sources of uncertainty in the 3d model, finding that variations in wind forcing had the largest impact on hypoxia estimates (J. P. Mattern, K. Fennel, and M. Dowd (2013), Sensitivity and Uncertainty Analysis of Model Hypoxia Estimates for the Texas-Louisiana Shelf, Journal of Geophysical Research, doi: 10.1002/jgrc.20130).

We agree that the mentioned manuscript provides a more elaborate analysis of uncertainty than Laurent and Fennel (2019), but not in a forecasting framework. We will add this reference to Line 55, for readers who are interested in exploring these uncertainties more.

L 77: After the first read, I am assuming that the DMO20 model has four compartments, an eastern and a western one, each divided into two layers. This could be made a bit more explicit in the text.

Thank you for the comment, we will add clarification in Section 2.1.

L 91: Is there any indication about the cause of this bias? It is nice to have an underlying parsimonious mechanistic model, yet the bias correction introduces a non-mechanistic element. By the way, it would be helpful to mention again that June is the start of the prediction interval and that the bias disappears over the course of several weeks, so that it can be neglected in the following month.

Thank you for the comment, the June bias might be due to multiple factors, including limitations of the DMO20 model formulation. We will add the suggested clarifications.

L 110: Is this done for one or multiple years? All years with data? This information is probably given later but it would be useful to mention it here already or even earlier.

Thank you for the comment, summer riverine inputs were sampled from multiple years, and we will add clarification.

L 118: Do the ten most relevant years represent the full time period accurately?

We appreciate the comment and the results of this study suggest that 10 relevant years perform well in representing the summer riverine inputs, which is reflected in increase of $R^2$ compared to randomly selected years (Table 2, compare overall for Cases 2 and 4).

Note that our goal is not to represent the entire study period, but to represent the range of summer conditions indicated by the flow and loading regressions (considering spring flows, rainfall, and loading). We recognize that there is some ambiguity regarding the appropriate number of relevant years to include. Thus, we have performed additional analysis (running the forecasts using only 5 relevant years) to test the sensitivity of the model to the number of relevant years included. We will add a brief discussion of this additional analysis to the manuscript.

L 144: It may be good to give some examples of the model parameters that contribute to the uncertainty here, so that the reader does not need to consult the DMO20 paper to get this information.

Thank you for the comment, and we are going to add the summary of the calibrated parameter estimates to the supplementary material.

L 150: What if one, multiple, or maybe all relevant years are in the future w.r.t. to the estimated year?

Thank you for the comment. Our approach assumes that there are sufficient records to generate a robust sample of representative years. Whether those years happen to be past or future is not directly relevant to our research questions. Clearly, there are sufficient samples moving forward from the present. If we wanted to generate a forecast in 1985, we'd need older records, but this is beyond the scope of this study, and we don't think it would be particularly meaningful to our research objectives.

L 156: "Sixteen multiple linear regressions": I assume, the 16 refers to 4 (months) * 2 (rivers) * 2 (discharge, nitrogen loading) but this could be made a bit clearer, or a reference to Table 1 could be added here already. In my opinion, it would be good to clearly state again that there are 4 regressions for each month.

We agree with the comment and will add this clarification.

L 185: The "hypoxia model" is DMO20, correct? I would suggest to include this here again.

We agree with the comment and will add this clarification.

L 192: Is there a distinction between "forecasted" in this line and "pseudo-forecast" a few lines above? It would be good to stay consistent with the use of "pseudo". Maybe even drop the "pseudo-" prefix after describing that this is the way the word forecast is used in the context of the manuscript.

Here, we use 'forecasted' as the adjective formed from a verb, while the product of forecasting in our case is 'pseudo-forecast' because we are generating the forecast post factum, as noted at Line 146.

Fig. 2: Am I correct in assuming that there are 32*30 red dots in the top panels, one for every day in June in the 32 years with data? But if the monitoring cruises are typically in late July, why are there so many red dots in the bottom panels?

The statement related to the top panels is correct. Red dots in the bottom panels indicate 34 available observations for June. There are total of 149 observations (i.e., cruises), 63 in July, 35 in August, 17 in September. Note that we are not just using LUMCON cruises. We are also using Texas A&M cruises, NOAA SEAMAP cruises, etc., so there are many cruises outside of July. See Matli et al., (2018) for more details. We will also clarify this in Section 2.2.

Fig. 4: I can only distinguish between 3 shades of gray here, yet the caption suggests there should be 4. Are the uncertainties plotted cumulatively or is the effect of parameter uncertainty generally smaller than that of the riverine and meteorological inputs?

There are four shades of grey. However, bands related to regression transformation of BWDO to HA (the lightest grey) do not add much visually to mechanistic model error, data input uncertainty and parameter uncertainty. These transformation regressions have high $R^2$ values and relatively small error ($R^2 = 0.98$ and the residual standard deviation is 706 km$^2$ for west section, while for east section the $R^2 = 0.99$ and the residual standard deviation is 216 km$^2$; see Del Giudice et al. (2020) for details). To address the reviewer's comment, we will switch the sequence of the shades of grey, so that the thin outer band is more prominent.

L 238: "Note that the relative magnitudes of the variance components are somewhat different from the relative magnitudes of the 95% IQR components ...": If the goal here is to say that the relative magnitudes differ because the variance has squared units I think it would be easier for the reader to state this directly, rather than drawing a line from IQR to standard deviation.

We agree with the comment and we will clarify the text.

**References**

Del Giudice, D., Matli, V. R. R. and Obenour, D. R.: Bayesian mechanistic modeling characterizes Gulf of Mexico hypoxia: 1968–2016 and future scenarios, Ecol. Appl., 30(2), eap.2032, doi:10.1002/eap.2032, 2020.
Laurent, A. and Fennel, K.: Time-Evolving, Spatially Explicit Forecasts of the Northern Gulf of Mexico Hypoxic Zone, Environ. Sci. Technol., 53(24), 14449–14458, doi:10.1021/acs.est.9b05790, 2019.
Matli, V. R. R., Fang, S., Guinness, J., Rabalais, N. N., Craig, J. K. and Obenour, D. R.: Space-Time Geostatistical Assessment of Hypoxia in the Northern Gulf of Mexico, Environ. Sci. Technol., 52(21), 12484–12493, doi:10.1021/acs.est.8b03474, 2018.

---

## Author Comment (AC2)

**Responses to comments on: "Daily hypoxia forecasting and uncertainty assessment via Bayesian mechanistic model for the Northern Gulf of Mexico" (Referee #2)**

Our responses are in blue.

**Comments**

This manuscript builds upon previous statistical models for hypoxia in the Gulf of Mexico to make a temporally resolved forecast. This work is potentially valuable for management of the ecosystem's fisheries and offers some important insights regarding the contributions of different sources of uncertainty to this type of forecast. The manuscript is well written and will be of interest to readers of HESS. I have one substantial concern and a number of comments for the authors to consider.

We appreciate the positive feedback and acknowledgement of potential utility to management.

This modeling approach is impressive as it handles the different sources of uncertainty without excessive parameterization or an actual process-based model. To do accomplish this, the model relies on use of historical data for the trajectory of summer meteorology, discharge and loading. The authors select the most relevant years' summer records based on similarity of spring forcings. This selection process at line 115 is explained clearly, but more justification would be helpful as this is potentially influential for the forecast. What are the consequences if this selection step were omitted? What happens to predictive performance if 10 less relevant years are used? Does that lead to degradation of performance?

Thank you for the positive comment on forecasting approach. A tighter selection (with fewer relevant years) may produce more accurate forecasting results, but may fail to capture the true variance (stochasticity) in the hydrometeorology. Thus, 10 years was selected as a reasonable balance between accuracy of prediction and accuracy of uncertainty characterization. To explore this more, we ran the forecasting procedure selecting only 5 relevant years, and the predictive accuracy was slightly improved (compare figure below to Figure 3 of manuscript). However, we estimate the confidence interval for the population variance to increase by a factor of 1.5 when using 5 years instead of 10 years. We will add a discussion of this analysis to the Discussion section of the manuscript.

If we omit the selection process entirely, performance degrades substantially. This is actually explored in Section 3.3. In particular, compare Case 4 to Case 2 in Table 2.

[Figure]

It appears that this temporally resolved forecast would, in fact, be static once the spring hydrology data are used to identify the most similar years to use for summer forcings. Could this somehow be informed by additional data from after May, perhaps using some information from previous years?

In general, we agree that it could be interesting to update the forecast over the summer, as additional data become available (after May). While outside the scope of this study, we will note this as potential future research in the Discussion. However, we are unsure what the reviewer means by "using some information from previous years", as we already do this. In addition, we will update the title of the manuscript to "Temporally resolved coastal hypoxia forecasting and uncertainty assessment via Bayesian mechanistic modeling" clarify the main point of this work.

Figure 2. Are the hindcasted data for all of the years, or only those that were matched as most relevant and used for the model?

This figure shows results for all years. Hindcasted data represents the predictive output of DMO20 using the actual hydrometeorology (assuming it is known throughout the summer). Thus, "relevant years" aren't relevant for hindcasts. The pseudo-forecasts are based on relevant years (and the relevant years change for each year being forecasted). We will edit the figure caption to clarify. Note that we will also edit Figure 1 to include both the hindcast and pseudo-forecast procedure, which should help to further clarify the methodology for the reader. The draft of the revised Figure 1 is shown below.

**(A) Hindcast**

[Figure]

**(B) Pseudo–forecast**

[Figure]

Figure 4. The predictive intervals in the shaded region do not appear to increase over the course of the season, which is unexpected based on Figure 3. The authors provide parts of the explanation for this in lines 240-245, but I suggest adding an additional sentence that puts together the 1) disparity in uncertainty between inputs and fitted parameters and 2) change in contribution of inputs over the season.

Thank you for the comment, and we agree that the predictive intervals in Figure 4 may seem unexpected. This is largely because predictive intervals tend to increase with hypoxic area size, and the hypoxic area size tends to decline after July. For this reason, we included Figure 5 and corresponding text at lines 258-268. As shown in Figure 5A, normalized predictive intervals (normalized by hypoxic area) do increase over the summer season (though they are still fairly high in mid June due to strong and highly variable wind speeds, as shown in Figure 5B). We will edit the text here to further clarify this point.

Table 2 and Line 326. The caution about increasing uncertainties over the season is appropriate, but is not adequately captured in the figures or tables in the main text or supplement. Instead, the decrease in R2 is presented. Although the uncertainties are referenced in the text, it would be

helpful to have a table or visual that shows the uncertainty for the pseudo-forecast by month (by that I mean IQR in units of HA or BWDO)

Thank you for the comment, however, we do not think the additional table is necessary as we discuss the increase of uncertainty in later August–September in the Lines 260-262. Besides, the uncertainty increase in later months can be seen in Figure 5. As noted in our previous response, we will further clarify this in the text.

**Technical Comments**

Table 2. What does the bold styling indicate? It appears to be the highest R2 of the four cases, but this could be explained in the caption

Thank you for the comment, the bold styling indeed indicates the highest $R^2$ of the four cases (by month) and we will add clarification in the table caption.

Line 91 – The rationale for this correction is clear and sound. Presumably there is some uncertainty associated with both the predictor and fitted parameters of this regression. Were those carried forward into the forecast? I would expect that uncertainty to be impactful.

We agree with the comment and most likely the uncertainty for early June forecasted values will increase. We will add the uncertainty associated with this regression to the model and update the manuscript. However, we do not expect it to considerably increase total forecasting uncertainty.

Line 135 – Did this occur frequently? Assigning a threshold for performance is appropriate, but consider adding some justification for why this particular threshold was selected.

Thank you for the comment. All the final regression models and their performance are presented in Table 1. As shown in Table 1, there is a large drop in performance between July and August (as noted in Lines 174-175), so results would be robust to moderate increases in this threshold. Also, we performed leave-one year-out cross validation (see response to Reviewer #1), which suggested even poorer performance of August–September regressions in projecting both flow and loading. We will update Table 1 with leave-one-out cross validation results and add discussion supporting our decision.

---

## Author Comment (AC3)

**Responses to comments on: "Daily hypoxia forecasting and uncertainty assessment via Bayesian mechanistic model for the Northern Gulf of Mexico" (Referee #3)**

Our responses are in blue.

**General comments**

This is a well written manuscript on an extremely important topic of predicting hypoxia in a water body of global significance. The methods and findings from this study are also applicable to similar water bodies across the world. The novel approach of using data-driven modelling (particularly, Bayesian statistical modelling) makes this a suitable paper for HESS. The results and discussion are generally well written (save one comment about being confused by Table 2).

We are grateful for the referee's positive comment and the recognition of contribution of this study.

However, I do think that it would be good to see more details in the methodology about the approach taken in this study. This is coming from the perspective of someone who thinks this work is very relevant to the work of watershed planners and managers – who may want more detail on how they could apply similar work themselves.

I understand the reasoning for not detailing the Bayesian Mechanistic Model in this paper (because it was previously published) - however, I do think that more details on the model formulation are required in Section 2.1 so that this can be a stand-alone paper in its own right. Particularly - what are the key parameters and what are the model equations? Similarly in Section 2.2, readers need to refer to the DMO 20 publication, and to Matli et al. 2018. I do think that these details should be brought across to this paper too.

Thank you for the comments and suggestions. In this manuscript, we focus on the new forecasting approach, but not the process-based model, which was described by DMO20. To address the reviewer's concern, we will add the main DMO20 model equations and a summary of the Bayesian posterior parameter estimates to the Supplementary material. We will also provide more explanation of Matli et. al., (2018) in the methods (Section 2.2).

I also think more emphasis in the methodology needs to be placed on the forecasting method and the regression modelling of June-September Discharge and Loading. I didn't realise until quite later in the paper that Bayesian methods were used in determining these data. This is quite important and could be applied not only in the context presented in the paper, but to other sites. In particular the details I would like to see are: what software/platform was used for the modelling (is this available on github etc?), how did you check convergence of chains?, how many iterations?, was there a burn-in period?, how many models did you produce in the exhaustive search (just to place the scale of the work in context), what are the prior distributions?, how were the assumptions checked?

We appreciate the comment and would like to point out that we use conventional linear regression (not Bayesian) to predict June–September discharge and loading. We preferred conventional over Bayesian methods due to the lack of strong prior information and higher

computational efficiency in the exhaustive search application. For situations without prior information, Bayesian and frequentist predictions are generally equivalent. We describe the process of constructing the regressions in Section 2.4 and results of the regressions in Section 3.1 and the observed vs predicted plots (Figures S3.2–S3.5). We did use the "Bayesian Information Criterion" for variable selection, but this is really more of a frequentist method, despite what the name might imply. We report the relevant R packages and methods in the Methods, and we will add a general reference to R (the software platform used in this study).

I would also like to see an assessment or further discussion of how the uncertainty in the forecasts are propogated through the Bayesian Mechanistic Model. How do you account for this? How does the final uncertainty change? More details on how this assessment is conducted would be good to see in the methodology.

Thank you for the comment. In order to improve the description of uncertainty propagation, we will update Figure 1 with symbols that indicate linkages where uncertainties are propagated through the forecasting procedure. In addition, we will add boxes for the regression conversions to Figure 1. Please see the draft of the updated Figure 1 below this response. We think Figure 4 and the associated discussion provides a useful assessment of the various uncertainties accounted for in the forecasting approach.

**Specific Comments:**

Lines 60-63: I am a little confused as to what the difference here is between hindcasting and forecasting? Would these need different modelling/simulation strategies? Is the forecasting process the same as the hindcasting process - just using input data representing future conditions? Then how would you validate the success of the forecasting? Perhaps a brief clarification of these points would be useful here.

Thank you for the comment. Hindcasts represent model predictions of past conditions assuming all input data are known throughout the prediction period (e.g., flows, loads, and winds are known throughout the summer). Forecasting, on the other hand, requires us to make predictions and/or assumptions regarding some model inputs (e.g., flows, loads, and winds after the forecast release date of 1 June). We explain the forecasting process in the Methods Section 2.3. To improve the differentiation between forecasting and hindcasting, we will expand Figure 1 to include a flowchart for hindcasting (see draft above). The most compelling validation of the pseudo-forecasts is comparison to the geostatistical "observations" (which are based on monitoring cruise sampling data). We also compare pseudo-forecasts to hindcasts, since hindcasts represent the best model estimate of historical hypoxic conditions.

Line 101: 'geostatistical estimates from Matli et al' - please elaborate on this a bit more: what are the geostatistical estimates of? What does Matli et al. 2018 provide?

Please see our response to the next comment.

Line 101-102: Could you please provide a few more details here too? I assume that this is referring to the estimates of BWDO and HA, but it is not 100% clear. Why is it that the monitoring cruise data have lower uncertainty? Also - how often do these monitoring cruises happen?

We will expand the description of these geostatistical estimates. In general, there are 149 cruises across our study period: 34 in June, 63 in July, 35 in August, and 17 in September. Geostatistical estimates are based on the cruise observations, such that there is greater uncertainty if/when trying to interpolate temporally (between cruises).

Line 321-324: The authors state that the projections of summer riverine inputs improve hypoxia forecasting skill - and refer to Table 2, however even after the explanation in Section 3.2, I am still confused about this table. I would have thought that if the projectings are improving forecasting skill, we should see higher R2 in the Forecasted vs Observed columns compared to the Forecasted vs Hindcasted columns. Or perhaps I am competely misunderstanding the table.

Thank you for this comment. Table 2 presents a comparison of using varying riverine and meteorological data inputs, as described earlier in the manuscript (Section 3.3). The purpose of Table 2 is to compare the four different forecasting input cases (focusing on a comparison across table rows, not columns). We will clarify this in the text by referencing Case 4 (best case) at Line 322. Forecast performance generally correlates more strongly with hindcasts than observations (because both forecasts and hindcasts are based on the same mechanistic hypoxia model). We also hope that our revised Figure 1 will help resolve some of this confusion.

**Technical Comments:**

 - The authors have used a large number of acronyms throughout the manuscript (e.g., HA, BDOM,  NGoM, MAR). This is perhaps a subjective comment, but I highly suggest that these acronyms are avoided and terms written out in full to make the manuscript more readable.

In the spirit of this comment, we will remove the NGoM and MAR acronyms.

- Fig S1: I suggest putting this figure in the main manuscript - not all readers are familiar with the Gulf of Mexico

Since there have been many publications on the northern Gulf hypoxic zone, we do not think it is critical to include this map in the main manuscript. However, if the editor agrees this is important, we will add it.

- figure 1 was a very useful figure for me and helped me understand the process - but for those who are not familiar with the different text box shapes - could you please provide a legend either in the figure or in the caption? also perhaps to make it easier for the reader, it might be nice to have the section heading numbers in the figure.

We appreciate the comment and will modify Figure 1 (see draft above). We have simplified the shapes, and we will also update the figure legend and/or caption.

**References**

Matli, V. R. R., Fang, S., Guinness, J., Rabalais, N. N., Craig, J. K. and Obenour, D. R.: Space-Time Geostatistical Assessment of Hypoxia in the Northern Gulf of Mexico, Environ. Sci. Technol., 52(21), 12484–12493, doi:10.1021/acs.est.8b03474, 2018.

---

## Author Response (AR1)

**Responses to comments on: "Daily hypoxia forecasting and uncertainty assessment via Bayesian mechanistic model for the Northern Gulf of Mexico" (Referee #1)**

Our responses are in blue. Line numbering in responses refer to the revised manuscript with changes incorporated.

**General comments**

One important piece of information that is not mentioned explicitly enough in the introduction or the abstract is that the main part of the manuscript is about using a statistical model to generate suitable parameters for an existing mechanistic model (referred to DMO20). Reading these sections for the second time, it becomes a bit more clear but it would be beneficial to the reader to describe this more clearly early in the manuscript.

Thank you for your feedback. We clarified in the introduction that we are using an existing mechanistic model with parameters that are probabilistically estimated through Bayesian inference (Line 59-61). Note that there is not a separate statistical model, per se. There is a mechanistic model calibrated using Bayesian inference.

The authors are careful in creating a forecasting scenario in which no summer data is included. However, the entire set of historical data is used to produce the linear regressions, which may include future data w.r.t. year for which the forecast is produced.

We appreciate the comment and understand the concern of the referee. Note that we do not use the predictions of these regressions directly in the forecast; we only apply them for defining the 10 relevant years (which cannot include the year being forecasted). To double-check the validity of these regressions, we performed leave-one year-out cross validation (LOOCV) where we excluded years by one, calibrated the models to the reduced dataset, and predicted for the excluded year. The performance of the August and September models dropped dramatically for cross-validated results compared to models built on the full dataset (see % change in the table below). However, as highlighted in the Results (Lines 191-193), we are only using the June and July regressions (highlighted with bold in a table below) in the proposed forecasting procedure (i.e., for selection of relevant years). We added the description of this additional analysis to the manuscript (Lines 152-154, 190-192) and expanded Table 1.

| model predictand | $R^2$ from Table 1 | LOOCV $R^2$ | % change |
|---|---|---|---|
| $\sqrt{Q_{A6}}$ | **0.79** | **0.75** | **-5.1** |
| $\sqrt{Q_{A7}}$ | **0.47** | **0.38** | **-19.1** |
| $\sqrt{Q_{A8}}$ | 0.28 | 0.14 | -50.0 |
| $\sqrt{Q_{A9}}$ | 0.13 | 0.04 | -69.2 |
| $\sqrt{L_{A6}}$ | **0.76** | **0.71** | **-6.6** |
| $\sqrt{L_{A7}}$ | **0.51** | **0.41** | **-19.6** |
| $\sqrt{L_{A8}}$ | 0.30 | 0.17 | -43.3 |

| | | | |
|---|---|---|---|
| $\sqrt{L_{A9}}$ | 0.11 | -0.02 | -118.2 |
| $\sqrt{Q_{M6}}$ | **0.77** | **0.73** | **-5.2** |
| $\sqrt{Q_{M7}}$ | **0.48** | **0.38** | **-20.8** |
| $\sqrt{Q_{M8}}$ | 0.28 | 0.15 | -46.4 |
| $\sqrt{Q_{M9}}$ | 0.13 | 0.04 | -69.2 |
| $\sqrt{L_{M6}}$ | **0.78** | **0.74** | **-5.1** |
| $\sqrt{L_{M7}}$ | **0.51** | **0.41** | **-19.6** |
| $\sqrt{L_{M8}}$ | 0.25 | 0.12 | -52.0 |
| $\sqrt{L_{M9}}$ | 0.09 | 0.01 | -88.9 |

If I understand the approach correctly, even in the Case 4 setup, the forecast may include information from summer data of the current year, if the current year is "relevant" (as defined in the manuscript). As a result, does the forecasting system produce significantly better results for the "relevant" years compared to other years? In addition to the 4 cases currently included in the manuscript, I would suggest to add a Case 5 that excludes all data from the future (forecasts for the first few years with little data could be skipped) or, alternatively, excludes the data from the current forecast year, even if it is relevant.

Actually, the pseudo-forecasts do not include the information from the summer of the current year (forecast year) in any of the forecasting scenarios, as stated in the Methods at Lines 156-157. In other words, when the procedure selects the 10 relevant years, the summer data for the forecast year is excluded from selection. We added an additional clarification at Lines 121-122. We also added a brief note to the Results (Section 3.2, Line 201) to remind readers of this important point. We do not think that the suggested Case 5 is warranted for this reason and because it does not directly address the study's research objectives (Lines 64-69).

We modified Figure 2 (which previously was Figure 1) (see draft below) to clarify that the forecasts only use observed data up until 31 May of the forecast year (since the nominal forecast release date is 1 June). Data for after 31 May are sampled only from "other years".

Looking at the results in Figure 2 and 3, it appears as if the forecast-hindcast as well as the forecast-observation comparisons show a pattern in August and September: The forecast appears to overestimate BWDO and HA for low values and underestimate it for high values and this pattern appears to increase in time. The authors already introduce a linear regression for the purpose of bias reduction but apply it only to the June forecast. A similar linear correction could be applied to correct this pattern which appears to increase with lead time. Yet, I am a bit hesitant to recommend such a correction because it, just like the bias reduction, adds a non-mechanistic element to the model.

Thank you for the comment, in fact we agree with the reviewer regarding some bias presence in September for the east section. However, we decided not to introduce a September bias correction for two main reasons. First, there are fewer measurements of hypoxia available for September, so the case for a bias correction is less strong (Figure S2.2). Second, we see that HA is predicted surprisingly well in September, even without a bias correction (Figure S2.3). We

also note that hypoxia is typically reduced in September compared to June–August, so September is of less interest to management and fisheries compared to other months.

[Figure]

How difficult would it be to extend the approach presented in this study and estimate August and September values from all data available until then? I am not suggesting that this needs to be done in this manuscript, yet creating successive 2 month forecasts appear a suitable course of action for producing more accurate estimates. This could be mentioned in the discussion.

We appreciate the comment and agree that being able to update and extend the forecast over the course of the summer could be beneficial for fisheries management. In this manuscript, we focus on a 1 June forecast release date, consisting with current Gulf forecasting practices. However, the methods should be transferable to other forecast release dates. We expanded the Discussion, pointing this out as a future opportunity (Lines 384-387).

Overall, while the manuscript is well written it sometimes overestimates the study-specific knowledge of the reader. Including more information explicitly would benefit readers. In some instances, I had to read ahead to answer questions which could have been addressed right away. I have listed some of those instances below.

Thank you for the positive feedback. In addition to addressing your specific comments, we reviewed and edited the entire manuscript for clarity.

**Specific comments**

L 8: "Several models" Here it would be helpful to specify what type of model is meant, e.g. "dynamical", "statistical" etc.

We appreciate the comment and added clarification to the abstract (Lines 8-9). Furthermore, we expand on the modeling approaches to predict and forecast hypoxia later in introduction (Lines 38-40, 47-55).

L 52: The same group has previously considered different sources of uncertainty in the 3d model, finding that variations in wind forcing had the largest impact on hypoxia estimates (J. P. Mattern, K. Fennel, and M. Dowd (2013), Sensitivity and Uncertainty Analysis of Model Hypoxia Estimates for the Texas-Louisiana Shelf, Journal of Geophysical Research, doi: 10.1002/jgrc.20130).

We agree that the mentioned manuscript provides a more elaborate analysis of uncertainty than Laurent and Fennel (2019), but not in a forecasting framework. We added this reference to Line 55, for readers who are interested in exploring these uncertainties more.

L 77: After the first read, I am assuming that the DMO20 model has four compartments, an eastern and a western one, each divided into two layers. This could be made a bit more explicit in the text.

Thank you for the comment, we added clarification in Section 2.1 (Lines 78-80).

L 91: Is there any indication about the cause of this bias? It is nice to have an underlying parsimonious mechanistic model, yet the bias correction introduces a non-mechanistic element. By the way, it would be helpful to mention again that June is the start of the prediction interval and that the bias disappears over the course of several weeks, so that it can be neglected in the following month.

Thank you for the comment, the June bias might be due to multiple factors, including limitations of the DMO20 model formulation. We added the suggested clarification (Lines 94-95).

L 110: Is this done for one or multiple years? All years with data? This information is probably given later but it would be useful to mention it here already or even earlier.

Thank you for the comment, summer riverine inputs were sampled from multiple years, and we added clarification (Line 121).

L 118: Do the ten most relevant years represent the full time period accurately?

We appreciate the comment and the results of this study suggest that 10 relevant years perform well in representing the summer riverine inputs, which is reflected in increase of $R^2$ compared to randomly selected years (Table 2, compare overall for Cases 2 and 4).

Note that our goal is not to represent the entire study period, but to represent the range of summer conditions indicated by the flow and loading regressions (considering spring flows, rainfall, and loading). We recognize that there is some ambiguity regarding the appropriate number of relevant years to include. Thus, we have performed additional analysis (running the forecasts using only 5 relevant years) to test the sensitivity of the model to the number of relevant years included. We added a brief discussion of this additional analysis to the manuscript (Lines 219-226).

L 144: It may be good to give some examples of the model parameters that contribute to the uncertainty here, so that the reader does not need to consult the DMO20 paper to get this information.

Thank you for the comment, and we added the summary of the calibrated parameter estimates to the supplementary material (Table S1) and referred to in in the main text (Lines 90-91).

L 150: What if one, multiple, or maybe all relevant years are in the future w.r.t. to the estimated year?

Thank you for the comment. Our approach assumes that there are sufficient records to generate a robust sample of representative years. Whether those years happen to be past or future is not directly relevant to our research questions. Clearly, there are sufficient samples moving forward from the present. If we wanted to generate a forecast in 1985, we'd need older records, but this is beyond the scope of this study, and we don't think it would be particularly meaningful to our research objectives.

L 156: "Sixteen multiple linear regressions": I assume, the 16 refers to 4 (months) * 2 (rivers) * 2 (discharge, nitrogen loading) but this could be made a bit clearer, or a reference to Table 1 could be added here already. In my opinion, it would be good to clearly state again that there are 4 regressions for each month.

We agree with the comment and changed the text for clarification (Lines 173-174).

L 185: The "hypoxia model" is DMO20, correct? I would suggest to include this here again.

We agree with the comment and added clarification (Line 202).

L 192: Is there a distinction between "forecasted" in this line and "pseudo-forecast" a few lines above? It would be good to stay consistent with the use of "pseudo". Maybe even drop the "pseudo-" prefix after describing that this is the way the word forecast is used in the context of the manuscript.

Here, we use 'forecasted' as the adjective formed from a verb, while the product of forecasting in our case is 'pseudo-forecast' because we are generating the forecast post factum, as noted at Lines 156-157.

Figure 2: Am I correct in assuming that there are 32*30 red dots in the top panels, one for every day in June in the 32 years with data? But if the monitoring cruises are typically in late July, why are there so many red dots in the bottom panels?

The statement related to the top panels is correct. Red dots in the bottom panels indicate 34 available observations for June. There are total of 149 observations (i.e., cruises), with 63 in July, 35 in August, 17 in September. Note that we are not just using LUMCON cruises. We are also using Texas A&M cruises, NOAA SEAMAP cruises, etc., so there are many cruises outside of July. See Matli et al., (2018) for more details. We also clarified this in Section 2.2 (Lines 108-113).

Figure 4: I can only distinguish between 3 shades of gray here, yet the caption suggests there should be 4. Are the uncertainties plotted cumulatively or is the effect of parameter uncertainty generally smaller than that of the riverine and meteorological inputs?

There are four shades of gray. However, bands related to regression transformation of BWDO to HA (the lightest gray) do not add much visually to mechanistic model error, data input uncertainty and parameter uncertainty. These transformation regressions have high $R^2$ values and relatively small error ($R^2 = 0.98$ and the residual standard deviation is 706 km$^2$ for west section, while for east section the $R^2 = 0.99$ and the residual standard deviation is 216 km$^2$; see Del Giudice et al. (2020) for details). To address the reviewer's comment, we switched the sequence of the shades of gray, so that the thin outer band is more prominent, see revised Figure 5.

L 238: "Note that the relative magnitudes of the variance components are somewhat different from the relative magnitudes of the 95% IQR components ...": If the goal here is to say that the relative magnitudes differ because the variance has squared units I think it would be easier for the reader to state this directly, rather than drawing a line from IQR to standard deviation.

We agree with the comment and we clarified the text (Line 265).

**Responses to comments on: "Daily hypoxia forecasting and uncertainty assessment via Bayesian mechanistic model for the Northern Gulf of Mexico" (Referee #2)**

Our responses are in blue. Line numbering in responses refer to the revised manuscript with changes incorporated.

**Comments**

This manuscript builds upon previous statistical models for hypoxia in the Gulf of Mexico to make a temporally resolved forecast. This work is potentially valuable for management of the ecosystem's fisheries and offers some important insights regarding the contributions of different sources of uncertainty to this type of forecast. The manuscript is well written and will be of interest to readers of HESS. I have one substantial concern and a number of comments for the authors to consider.

We appreciate the positive feedback and acknowledgement of potential utility to management.

This modeling approach is impressive as it handles the different sources of uncertainty without excessive parameterization or an actual process-based model. To do accomplish this, the model relies on use of historical data for the trajectory of summer meteorology, discharge and loading. The authors select the most relevant years' summer records based on similarity of spring forcings. This selection process at line 115 is explained clearly, but more justification would be helpful as this is potentially influential for the forecast. What are the consequences if this selection step were omitted? What happens to predictive performance if 10 less relevant years are used? Does that lead to degradation of performance?

Thank you for the positive comment on forecasting approach. A tighter selection (with fewer relevant years) may produce more accurate forecasting results, but may fail to capture the true variance (stochasticity) in the hydro-meteorology. Thus, 10 years was selected as a reasonable balance between accuracy of prediction and accuracy of uncertainty characterization. To explore this more, we ran the forecasting procedure selecting only 5 relevant years, and the predictive accuracy was slightly improved (compare figure below to Figure 4 of manuscript). However, we estimate the confidence interval for the population variance to increase by a factor of 1.5 when using 5 years instead of 10 years. We added a discussion of this analysis to the Discussion section of the manuscript (Lines 219-226).

If we omit the selection process entirely, performance degrades substantially. This is actually explored in Section 3.3. In particular, compare Case 4 to Case 2 in Table 2.

[Figure]

It appears that this temporally resolved forecast would, in fact, be static once the spring hydrology data are used to identify the most similar years to use for summer forcings. Could this somehow be informed by additional data from after May, perhaps using some information from previous years?

In general, we agree that it could be interesting to update the forecast over the summer, as additional data become available (after May). While outside the scope of this study, we noted this as potential future research in the Discussion (Line 384-387). However, we are unsure what the reviewer means by "using some information from previous years", as we already do this. In addition, to avoid the possible confusion about the forecast delivery frequency and timeframe, we have changed the title of the manuscript to "Temporally resolved coastal hypoxia forecasting and uncertainty assessment via Bayesian mechanistic modelling"

Figure 2. Are the hindcasted data for all of the years, or only those that were matched as most relevant and used for the model?

This figure shows results for all years. Hindcasted data represents the predictive output of DMO20 using the actual hydro-meteorology (assuming it is known throughout the summer), which we additionally clarify in Lines 89-91. Thus, "relevant years" aren't relevant for hindcasts. The pseudo-forecasts are based on relevant years (and the relevant years change for each year being forecasted). We edited the figure caption to clarify. Note that we also revised Figure 2 (previously Figure 1) to include both the hindcast and pseudo-forecast procedure, which should help to further clarify the methodology for the reader. The revised Figure 2 is shown below.

**(A) Hindcast**

**(B) Pseudo–forecast**

Figure 4. The predictive intervals in the shaded region do not appear to increase over the course of the season, which is unexpected based on Figure 3. The authors provide parts of the explanation for this in lines 240-245, but I suggest adding an additional sentence that puts together the 1) disparity in uncertainty between inputs and fitted parameters and 2) change in contribution of inputs over the season.

Thank you for the comment, and we agree that the predictive intervals in Figure 5 (previously Figure 4) may seem unexpected. This is largely because predictive intervals tend to increase with increasing hypoxic area size, and the hypoxic area size tends to decline after midsummer. For this reason, we included Figure 6 (previously Figure 5) and corresponding text at Lines 285-296. As shown in Figure 6A, normalized predictive intervals (normalized by hypoxic area) do increase over the summer season (though they are still fairly high in mid June due to strong and highly variable wind speeds, as shown in Figure 6B). We edited the text here to further clarify this point (Lines 283-285).

Table 2 and Line 326. The caution about increasing uncertainties over the season is appropriate, but is not adequately captured in the figures or tables in the main text or supplement. Instead, the decrease in R2 is presented. Although the uncertainties are referenced in the text, it would be

helpful to have a table or visual that shows the uncertainty for the pseudo-forecast by month (by that I mean IQR in units of HA or BWDO)

Thank you for the comment, however, we do not think the additional table is necessary as we discuss the increase of uncertainty in later August–September in the Lines 295-296. Besides, the uncertainty increase in later months can be seen in Figure 6. As noted in our previous response, we clarified those points in the text.

**Technical Comments**

Table 2. What does the bold styling indicate? It appears to be the highest R2 of the four cases, but this could be explained in the caption

Thank you for the comment, the bold styling indeed indicates the highest $R^2$ of the four cases (by month) and we added clarification in the table caption.

Line 91 – The rationale for this correction is clear and sound. Presumably there is some uncertainty associated with both the predictor and fitted parameters of this regression. Were those carried forward into the forecast? I would expect that uncertainty to be impactful.

We agree with the comment and most likely the uncertainty for early June forecasted values will increase. We added the uncertainty associated with this regression to the model and updated the manuscript. Because these regressions fit the data very well ($R^{2\geq}0.98$, Section S1), the total forecasting uncertainty did not increase substantially. However, we modified the symbology in Figure 5 to make this additional uncertainty more visible.

Line 135 – Did this occur frequently? Assigning a threshold for performance is appropriate, but consider adding some justification for why this particular threshold was selected.

Thank you for the comment. All the final regression models and their performance are presented in Table 1. As shown in Table 1, there is a large drop in performance between July and August (as noted in Lines 190-192), so results would be robust to moderate increases in this threshold. Also, we performed leave-one-year-out cross validation (see response to Reviewer #1), which suggested even poorer performance of August–September regressions in projecting both flow and loading. We updated Table 1 with leave-one-out cross validation results and added discussion supporting our decision (Lines190-194).

**Responses to comments on: "Daily hypoxia forecasting and uncertainty assessment via Bayesian mechanistic model for the Northern Gulf of Mexico" (Referee #3)**

Our responses are in blue. Line numbering in responses refer to the revised manuscript with changes incorporated.

**General comments**

This is a well written manuscript on an extremely important topic of predicting hypoxia in a water body of global significance. The methods and findings from this study are also applicable to similar water bodies across the world. The novel approach of using data-driven modelling (particularly, Bayesian statistical modelling) makes this a suitable paper for HESS. The results and discussion are generally well written (save one comment about being confused by Table 2).

We are grateful for the referee's positive comment and the recognition of contribution of this study.

However, I do think that it would be good to see more details in the methodology about the approach taken in this study. This is coming from the perspective of someone who thinks this work is very relevant to the work of watershed planners and managers – who may want more detail on how they could apply similar work themselves.

I understand the reasoning for not detailing the Bayesian Mechanistic Model in this paper (because it was previously published) - however, I do think that more details on the model formulation are required in Section 2.1 so that this can be a stand-alone paper in its own right. Particularly - what are the key parameters and what are the model equations? Similarly in Section 2.2, readers need to refer to the DMO 20 publication, and to Matli et al. 2018. I do think that these details should be brought across to this paper too.

Thank you for the comments and suggestions. In this manuscript, we focus on the new forecasting approach, but not the process-based model, which was described by DMO20. To address the reviewer's concern, we added the main DMO20 model equations and a summary of the Bayesian posterior parameter estimates to the Supplementary material (Section S1). We also provided more explanation of Matli et. al. (2018) in the methods (Section 2.2, Lines 108-113).

I also think more emphasis in the methodology needs to be placed on the forecasting method and the regression modelling of June-September Discharge and Loading. I didn't realise until quite later in the paper that Bayesian methods were used in determining these data. This is quite important and could be applied not only in the context presented in the paper, but to other sites. In particular the details I would like to see are: what software/platform was used for the modelling (is this available on github etc?), how did you check convergence of chains?, how many iterations?, was there a burn-in period?, how many models did you produce in the exhaustive search (just to place the scale of the work in context), what are the prior distributions?, how were the assumptions checked?

We appreciate the comment and would like to point out that we use conventional linear regression (not Bayesian) to predict June–September discharge and loading. We preferred

conventional over Bayesian methods due to the lack of strong prior information and higher computational efficiency in the exhaustive search application. For situations without prior information, Bayesian and frequentist predictions are effectively equivalent. We describe the process of constructing the regressions in Section 2.4 and results of the regressions in Section 3.1 and the observed vs predicted plots (Figures S3.1–S3.4). We did use the "Bayesian Information Criterion" for variable selection, but this is really more of a frequentist method, despite what the name might imply. We report the relevant R package and methods for variables selection in the Methods (Lines 147-148), and we added a general reference to R (the software platform used in this study) (Lines 116-117).

I would also like to see an assessment or further discussion of how the uncertainty in the forecasts are propogated through the Bayesian Mechanistic Model. How do you account for this? How does the final uncertainty change? More details on how this assessment is conducted would be good to see in the methodology.

Thank you for the comment. In order to improve the description of uncertainty propagation, we updated Figure 2 (previously Figure 1) with symbols that indicate linkages where uncertainties are propagated through the forecasting procedure. In addition, we added boxes for the regression conversions to Figure 2. Please see the updated Figure 2 below this response. We think Figure 5 and the associated discussion provides a useful assessment of the various uncertainties accounted for in the forecasting approach.

**(A) Hindcast**

**(B) Pseudo–forecast**

**Specific Comments:**

Lines 60-63: I am a little confused as to what the difference here is between hindcasting and forecasting? Would these need different modelling/simulation strategies? Is the forecasting process the same as the hindcasting process - just using input data representing future conditions? Then how would you validate the success of the forecasting? Perhaps a brief clarification of these points would be useful here.

Thank you for the comment. Hindcasts represent model predictions of past conditions assuming all input data are known throughout the prediction period (e.g., flows, loads, and winds are known throughout the summer), which is now clarified at Lines 89-91. Forecasting, on the other hand, requires us to make predictions and/or assumptions regarding some model inputs (e.g., flows, loads, and winds after the forecast release date of 1 June). We explain the forecasting process in the Methods, Section 2.3. To improve the differentiation between forecasting and hindcasting, we expanded Figure 2 (previously Figure 1) to include a separate flowchart for hindcasting (see above). The most compelling validation of the pseudo-forecasts is comparison to the geostatistical "observations" (which are based on monitoring cruise sampling data). We also compare pseudo-forecasts to hindcasts, since hindcasts represent the best model estimate of historical hypoxic conditions.

Line 101: 'geostatistical estimates from Matli et al' - please elaborate on this a bit more: what are the geostatistical estimates of? What does Matli et al. 2018 provide?

Please see our response to the next comment.

Line 101-102: Could you please provide a few more details here too? I assume that this is referring to the estimates of BWDO and HA, but it is not 100% clear. Why is it that the monitoring cruise data have lower uncertainty? Also - how often do these monitoring cruises happen?

We expanded the description of these geostatistical estimates (Lines 108-113). In general, there are 149 cruises across our study period: 34 in June, 63 in July, 35 in August, and 17 in September. Geostatistical estimates are based largely on the cruise observations, such that there is greater uncertainty if/when trying to interpolate temporally (between cruises).

Line 321-324: The authors state that the projections of summer riverine inputs improve hypoxia forecasting skill - and refer to Table 2, however even after the explanation in Section 3.2, I am still confused about this table. I would have thought that if the projectings are improving forecasting skill, we should see higher R2 in the Forecasted vs Observed columns compared to the Forecasted vs Hindcasted columns. Or perhaps I am competely misunderstanding the table.

Thank you for this comment. Table 2 presents a comparison of using varying riverine and meteorological data inputs, as described earlier in the manuscript (Section 3.3). The purpose of Table 2 is to compare the four different forecasting input cases (focusing on a comparison across table rows, not columns). We have additionally clarified this in the text by referencing Case 4 (best case) at Line 352. Forecast performance generally correlates more strongly with hindcasts than observations (because both forecasts and hindcasts are based on the same mechanistic hypoxia model; see Lines 238-240). We also hope that our revised flowchart (Figure 2) will help resolve some of this confusion.

**Technical Comments:**

 - The authors have used a large number of acronyms throughout the manuscript (e.g., HA, BDOM,  NGoM, MAR). This is perhaps a subjective comment, but I highly suggest that these acronyms are avoided and terms written out in full to make the manuscript more readable.

In the spirit of this comment, we have removed the NGoM and MAR acronyms.

- Fig S1: I suggest putting this figure in the main manuscript - not all readers are familiar with the Gulf of Mexico

We have combined Figure S1 and Figure S3.1 into Figure 1 representing the study site and the watershed area and placed it in the main text of the manuscript.

- figure 1 was a very useful figure for me and helped me understand the process - but for those who are not familiar with the different text box shapes - could you please provide a legend either

in the figure or in the caption? also perhaps to make it easier for the reader, it might be nice to have the section heading numbers in the figure.

We appreciate the comment and modified Figure 2 (see above). We have simplified the shapes, and we also updated the figure caption.

**References**

Del Giudice, D., Matli, V. R. R. and Obenour, D. R.: Bayesian mechanistic modeling characterizes Gulf of Mexico hypoxia: 1968–2016 and future scenarios, Ecol. Appl., 30(2), eap.2032, doi:10.1002/eap.2032, 2020.

Laurent, A. and Fennel, K.: Time-Evolving, Spatially Explicit Forecasts of the Northern Gulf of Mexico Hypoxic Zone, Environ. Sci. Technol., 53(24), 14449–14458, doi:10.1021/acs.est.9b05790, 2019.

Matli, V. R. R., Fang, S., Guinness, J., Rabalais, N. N., Craig, J. K. and Obenour, D. R.: Space-Time Geostatistical Assessment of Hypoxia in the Northern Gulf of Mexico, Environ. Sci. Technol., 52(21), 12484–12493, doi:10.1021/acs.est.8b03474, 2018.

---

## Author Response (AR2)

**Responses to comments on: "Temporally resolved coastal hypoxia forecasting and uncertainty assessment via Bayesian mechanistic modelling" (Referee #1)**

Our responses are in blue. Line numbering in responses refer to the revised manuscript with changes incorporated.

**General comments**

The authors addressed all of my comments and even included a leave-one out cross validation in the revised manuscript, with some interesting, and in my view, valuable results. I think these results should be given a bit more room in the manuscript, beyond the current 2 sentences, which just mention that the LOOCV was performed, and that one of its results confirmed those of the calibration. I would suggest including a bit of motivation and at least the information included in the response to the comment in my previous review.

Thank you for your feedback. We added a sentence motivating the LOOCV and a relevant reference to the Methods (Lines 155-156). We also enhanced the description in the Results (Lines 192-194).

**Specific comments**

l 94: "Predictions of BWDO and HA [...] are hereinafter referred to as "hindcasts"": The term "predictions" makes it sound more like a forecast, I would recommend to use "estimates" instead. Similarly, in the next sentence, "estimated" could be used instead of "predicted".

We appreciate the comment and modified the text as suggested (Lines 89, 93).

l 151: "river (Atchafalaya and Mississippi) discharge (Q_A and Q_M, [...])": I would suggest merging the terms in parentheses: "river discharge (Q_A for the Atchafalaya and Q_M for the Mississippi, [...])"

We agree with the comment and changed the text as suggested (Line 143).

l 154: I think this should now be a reference to Fig 2.

Thank you for the comment. This is a reference to the map of the river basin (Fig 1). However, since we have moved this map to the main manuscript, the reference is less critical, and we remove it to avoid confusion (Line 146).

l 170: "parameter, model residual, transformation, and bias adjustment uncertainties": I think, it would be clearer to write "uncertainties in parameters, model residuals, transformation, and bias adjustment", but I would also recommend adding which transformation is meant here.

We agree with the comment and we clarified the text (Line 163)

Table 1: Reference LOOCV R^2 in the caption.

We appreciate the comment and expanded the table caption, providing explanation for LOOCV (Lines 200-202).

l 234: "$R^2$ increases by 2.8%": It would be useful here to specify if this refers to percentage points or percent change relative to the base value; or maybe use "$R^2$ increases to x%". Previously, two $R^2$ were mentioned (west and east), does the 2.8% apply to both?

We appreciate the comment and clarified the text (Line 226). These values are based on the prediction of total shelfwide HA (also clarified at Line 211).

l 277: "Note that the relative magnitudes of the variance components are somewhat different..." This statement is referring to a figure reference that is in the supplemental material, and the reader will only understand or need it when seeing the figure. I would suggest moving the figure into the main manuscript, or add the note to the figure caption.

We agree with the comment and moved the sentence to supplemental material, and more specifically to the caption of Figure S4.2.

Fig. 4: I can more clearly see the darkest shade of gray in this plot, this is great. It would be nice to include a legend so that readers can reference the colors more easily.

Thank you for the comment. We added the legend to Figure 4.

Fig. 5: It would be useful to widen the figure, so that the individual boxes can be seen.

We appreciate the comment and increased the width of Figure 5.

**Responses to comments on: "Temporally resolved coastal hypoxia forecasting and uncertainty assessment via Bayesian mechanistic modelling" (Referee #2)**

Our responses are in blue. Line numbering in responses refer to the revised manuscript with changes incorporated.

**Comments**

The authors addressed all of the points that I raised in the first round of review. Figure 1 is helpful and I suggest adding units to the bathymetric lines or explaining those in the figure legend.

We appreciate the positive feedback. We clarified the isobath representation in the caption of Figure 1 (Lines 102-103).

**Responses to comments on: "Daily hypoxia forecasting and uncertainty assessment via Bayesian mechanistic model for the Northern Gulf of Mexico" (Referee #3)**

Our responses are in blue. Line numbering in responses refer to the revised manuscript with changes incorporated.

**General comments**

This is a valuable contribution in the field of predicting hypoxia.

We appreciate for the positive feedback on this study's contribution.